# Experimental Study on Subgrade Material of Calcium Silicate Slag

**DOI:** 10.3390/ma15062304

**Published:** 2022-03-20

**Authors:** De Zhang, Zhijie Yang, Dong Kang, Chenyang Fang, Yang Jiao, Shizhong Mi

**Affiliations:** 1School of Mining and Technology, Inner Mongolia University of Technology, Hohhot 010051, China; zhangde971026@163.com (D.Z.); 18538888088@163.com (D.K.); 20211100472@imut.edu.cn (C.F.); 20211800659@imut.edu.cn (Y.J.); 2The Key Laboratory of Green Development for Mineral Resources, Inner Mongolia University of Technology, Hohhot 010051, China; 3Inner Mongolia Road and Bridge Group Co., Ltd., Hohhot 010051, China; msz220203@163.com

**Keywords:** calcium silicate slag, inorganic binding stable materials, pavement base, micromorphology

## Abstract

Calcium silicate slag (CSS) is used as a secondary solid waste produced by aluminum extraction technology from high alumina fly ash, and its resource utilization has always been a key issue to be solved. In this study, CSS was used to replace a portion of fly ash (FA) to prepare a new inorganic binder stabilized material for road base. The unconfined compressive strength (UCS), phase composition, microstructure, durability and performance index of the base of the test section of the CSS pavement base material were studied. The results showed that with the increase in CSS content, the UCS of pavement base materials gradually increased. Under standard curing conditions, the UCS increased 6.90~17.24% after 7 days, and 7.90~28.95% after 28 days. The main reason was that as the hydration time increased from 7 d to 28 d, the hydration products C-A-S-H gel and C-S-H gel increased, the [SiO_4_] polymerization degree increased, the crystal type changed, and the structure denser, which supported the good development of mechanical strength of CSS pavement base material. In addition, the research has been successfully applied to a pilot test in Hohhot, China. The freeze–thaw resistance, water stability and UCS of the CSS pavement base material were tested to meet the requirements of Chinese road construction standards, indicating that the application of CSS in pavement base is feasible.

## 1. Introduction

Inorganic binding material stabilized base is attracting considerable critical attention. Cement stabilized base is the most commonly used type of pavement base used in construction. However, when cement is used to build road base, due to the influence of clay minerals, the hydration is not able to be completely reacted and, therefore, the optimal mechanical properties cannot be achieved [1]. In addition, CO_2_ is produced during cement production [2,3,4,5], which pollutes the environment. Replacing 10% of cement with low carbon materials can not only reduce carbon dioxide emissions by 6.1 million tons per year, but, in addition, the supplement of calcium to cementitious materials in inorganic binders can improve the alkalinity and mechanical properties of the mixture [6]. Therefore, in the application of road engineering, fly ash [7,8], high aluminum slag [9,10], and waste aggregate [11,12] have been considered as partial substitutes for cement.

Calcium silicate slag is the secondary solid waste generated in the process of extracting alumina from high alumina fly ash, with 2.2~2.5 tons of calcium silicate slag produced per ton of Al_2_O_3_ alumina, and 2.5 tons of fly ash consumed [13]. Long term CSS accumulation not only wastes a significant amount of land resources and causes air pollution, but, additionally, because it becomes alkaline itself, it will gradually alkalize the surrounding land in a humid environment, causing the loss of soil nutrients and polluting groundwater resources [14]. According to the physical and chemical properties of CSS and fly ash (FA), the authors carried out the development of CSS pavement base material with CSS as the main material. The resource utilization of CSS has become a hot research issue in recent years. Recently, many studies have been conducted on the utilization of CSS in the laboratory. Yan et al. [15] carried out research on CSS as a cement blending cementitious material, and found that the main hydration products of calcium silica slag cement clinker were amorphous C-(A)-S-H, Ca(OH)_2_ and a small amount of Aft, and compressive strength was higher than 32.5 MPa when the amount of CSS did not exceed 50%. Yang et al. [16] conducted a study on the hydration mechanism of ternary composite geopolymer cementitious materials prepared synergistically using CSS and FA slag; with the increase in FA/CSS, β-C_2_S in CSS hydrates formed Ca(OH)_2_ and C-S-H minerals, and polymerized with Si-O bonds and Al-O in fly ash and slag under an alkaline environment to produce C(N)-A-S-H gels. Bai et al. [17] showed that CSS promoted the degree of hydration of cement, and that the hydration of β-C_2_S to produce Ca(OH)_2_ decreased with the increasing substitution of CSS. Shi et al. [18] demonstrated the use of calcium silicate slag and blast furnace slag to prepare cementitious materials, and that the compressive strength could reach more than 65 MPa when CSS was dosed at 30% and cured for 28 d. Zhang et al. [19] studied the hydration of cementitious materials based on CSS. Their results showed that as the CSS replacement rate increased, the chemically bound water decreased and the spherical gel observed in the microstructure decreased, which improved the degree of hydration and reduced the heat of hydration.

To date, there has been research on the independent use of CSS as the main raw material of composite cementitious material. which also provides a theoretical basis for the synergistic preparation of composite cementitious materials from CSS. However, there has been little research on the application of inorganic binder CSS stabilized macadam semi-rigid subgrade, and the study of its mechanical properties, technical properties and microstructure needs further improvement. Therefore, this paper aims to study the preparation of a calcium silicate slag based road base material with different proportions of inorganic binders, with a main focus on the mechanical properties, durability, microstructure and hydration mechanism of the road base material. Meanwhile, the morphology and structure are characterized by XRD (Rigaku, Hitachi, Japan), FTIR (Shimadzu, Shanghai, China), SEM (Hitachi, Tokyo, Japan) and other different analytical methods. The research results expand the selection range of road base materials, provide an effective way for the treatment of CSS, reduce the processing cost, promote the circular development of the green low carbon economy, and reduce the project cost. This study is of great significance for the first application of inorganic binder calcium silicate slag stabilized macadam semi-rigid base in China.

## 2. Materials and Methods

### 2.1. Materials

The main raw materials for the test were CSS, FA and cement. The CSS was taken from the world’s first FA aluminum extraction production line, Datang International Renewable Resources Development company (Hohhot, China). The FA was produced from the world’s largest thermal power producer, Inner Mongolia Datang International Tuoketuo Power company (Hohhot, China). The cement was commercially available ordinary silicate P·O 42.5 cement. The chemical composition of CSS and FA is shown in Table 1. It can be seen from Table 1 that the content of CaO in the fly ash was less than 10%, belonging to class F. The obtained CSS and FA were dried in an oven at 105 °C for 3 h. The particle size of CSS was measured as 4.59 µm, the specific surface area was 0.575 m^2^/cm^3^, 80% of the particles were distributed between 1.5–9.5 µm, the median particle size of fly ash was 12.72 µm, the specific surface area was 0.510 m^2^/cm^3^. The CSS ball mill was finely ground for 20 min, and the measured particle size distribution is shown in Figure 1c. The XRD (Rigaku, Hitachi, Japan) patterns of FA and CSS are shown in Figure 1a,b. The results showed that the main mineral phase was β-dicalcium silicate (β-2CaO·SiO_2_). The main mineral phase of FA was glass phase, and the other parts were mullite (3Al_2_O_3_·2SiO_2_) and quartz (SiO_2_).

### 2.2. Experimental Method

The above-milled CSS, untreated FA and cements were used in the experiments. To investigate the effect of CSS on pavement base materials, proportions were designed with different percentages of CSS in inorganic binding materials, to obtain better mechanical properties. The raw material composition design is shown in Table 2.

Preparation of cylindrical specimens with diameter × height = Φ150 mm × 150 mm was conducted according to the Test Procedure of Inorganic Binding Material Stabilization for Highway Engineering (JTGE51-2009) [20]. The optimum moisture content was set according to the best moisture content obtained from compaction tests. The ratio of mixed of raw materials and inorganic binding material to stabilized gravel was 15:85. A constant temperature and humidity was maintained using YH-40B cement standard maintenance box material, maintenance temperature control at 20 ± 2 °C, and humidity control above 95%. The WHY-300 (Yixuan, Baoding, China) electro-hydraulic pressure tester was used to measure lateral limitless compressive strength after 6 d of maintenance of the specimens in fresh water (20 ± 1 °C) after 1 d of immersion. The measurement accuracy of the press was ±1%, and the loading speed of the press was effectively controlled at 1 mm/min. The flow chart of the test is shown in Figure 2. The hydration characteristics of CSS pavement subgrade materials without aggregate. The moisture content of the sample was the best moisture content, under the same conditions, the core of the net slurry specimens crushed by standard maintenance 7 d and 28 d were put into anhydrous ethanol to terminate the hydration for 24 h and then put into DHG-9620A (Zuole, Shanghai, China) type blast drying oven at 45 °C for 24 h. The material grinding was carried out by ball mill, and ground into powder after passing through a 200 mesh sieve for testing.

The scanning speed was 5°/min, the scanning range was 10–70°, the step was 0.02° and a copper target was used by the UltimalV X-ray diffraction (Rigaku, Hitachi, Japan) analyzer of Japanese Science. The Shimadzu Fourier infrared spectrometer IRTracer-100 (Shimadzu, Shanghai, China) was used for the IR test, with a wavelength range of 4000–400 cm^−1^ and a resolution of 4 cm^−1^. A HITACHI-3400N scanning electron microscope (Hitachi, Tokyo, Japan) was used for the microscopic morphology test. The SE resolution was ≤3 nm, BSE resolution was ≤4 nm, and acceleration voltage was 0.3–30 kV.

## 3. Results and Discussion

### 3.1. Unconfined Compressive Strength Test

In order to verify the alkali excitation effect of CSS content on active substances in FA materials, CSS and FA were mixed with ordinary Portland cement in to different proportions. The content of CSS increased from 0% to 13%, and the content of FA decreased from 13% to 0%. The unconfined compressive strength (UCS) standard tests were carried out at 7 d and 28 d. 

The UCS standard test results are shown in Figure 3. The 7 d UCS of CFC-1, CFC-2, CFC-3, CFC-4, CFC-5, CFC-6, and CFC-7 were 5.6 MPa, 5.9 MPa, 5.6 MPa, 5.5 MPa, 5.3 MPa, 5.3 MPa, and 5.1 MPa, respectively, and the 28 d UCS of CFC-1, CFC-2, CFC-3, CFC-4, CFC-5, CFC-6, and CFC-7 were 8.3 MPa, 8.7 MPa, 8.4 MPa, 8.3 MPa, 8.1 MPa, 8 MPa, 7.9 MPa, respectively. This data provides strong evidence that the UCS increased gradually with an increasing amount of CSS, with an increase in curing age. The growth trend of unconfined compressive strength was more obvious. The UCS of the cement stabilized gravel material reached 5.5 MPa at 7 d of maintenance when the amount of silica-calcium slag was 11%, which is in line with the standard specification of Chinese first-class highway pavement base. The UCS of cement stable gravel material with the addition of CSS was clearly higher than the unconfined compressive strength of cement stable gravel material without the addition of calcium silica slag, when the amount of calcium silica slag admixture was as high as 7%. This was mainly due to the higher alkali content in the silica-calcium slag, which improved the hydration of cement in a highly alkaline environment and induced the alkali excitation reaction of fly ash to generate a three-dimensional reticulated zeolite-like structure, which improved the early strength of the cement. 

The UCS curves at the different curing ages of 7 d and 28 d show that the advantage of CSS in improving the strength of cement stabilized aggregates gradually emerged with the increase in curing age. The UCS of CFC-2 was higher than CFC-7 by 0.6 MPa at a curing age of 7 d. The UCS of CFC-1 was significantly lower than that of CFC-2 due to the decrease in the proportion of fly ash in the inorganic binder, which led to the insufficient participation of [SiO_4_] and [AlO_4_] in the precursors of the hydration reaction, resulting in the generation of fewer hydration products [21]. The mechanical properties of CSS cement stabilized aggregates were significantly better than those of cement stabilized aggregates without CSS, when the amount of CSS was mixed into the range of 3~13% of the total binding material at 28 d.

### 3.2. XRD Results

Figure 4 illustrates the XRD analysis of different calcium silica slag base materials after 7 and 28 d of curing, while the hydrated synthetic mineral information of calcium silica slag pavement base materials is shown in Table 3. Figure 4a reveals a significant increase in the intensity of the diffraction peak angle of 29.47° for calcium carbonate, as shown in Equations (1) and (2), and CSH diffraction peaks. This is due to the silica in fly ash being mostly of glass phase structure, the material in the cement hydration producing Ca(OH)_2_, and the fly ash alkali excitation reaction changing its internal SiO_2_ into a crystal phase, to participate in the reaction into C-(A)-S-H, as shown in Equation (3).
(1)$$2\left(2\mathrm{CaO}\cdot {\mathrm{SiO}}_{2}\right)+4H_{2}O\to 3\mathrm{Cao}\cdot 2{\mathrm{SiO}}_{2}\cdot 3H_{2}O+\mathrm{Ca}{\left(\mathrm{OH}\right)}_{2}$$
(2)$$\mathrm{Ca}{\left(\mathrm{OH}\right)}_{2}+{\mathrm{CO}}_{2}\to \mathrm{Ca}{\mathrm{CO}}_{3}+H_{2}O$$
(3)$$\mathrm{Ca}{\left(\mathrm{OH}\right)}_{2}+{\mathrm{SiO}}_{2}+H_{2}O\to \mathrm{CaO}\cdot {\mathrm{SiO}}_{2}.2H_{2}O$$

Figure 4a indicates that the diffraction peak intensities of AFt and hydrated C-S-H gels were more significant in their XRD patterns under different CSS doping, in addition to β-C_2_S, calcite and other minerals introduced by CSS. Moreover, the XRD of calcium aluminate peak and C-S-H diffraction peak intensity of CSS with 11% dosing and 2% fly ash with 0%, 13% and 9% CSS hydration specimens were significantly enhanced under different calcium silicate slag dosing, which indicates that the CSS clinker with 11% dosing and 2% FA helped to improve the strength of CSS cement due to the volcanic ash reaction in the alkaline environment. The main reaction equations are shown in Equations (4)–(9) [22]. As indicated by the volcanic ash reaction, the reactive silica and alumina in fly ash reacted with the Ca(OH)_2_ produced by the preliminary hydration reaction. Equations (10)–(12) [22] indicate that the two reactions were carried out simultaneously, the C-S-H content varied continuously with the substitution rate of calcium silica slag, the cement dosing indicated that the two reactions were carried out simultaneously and the C-S-H content varied continuously with the substitution rate of CSS and the cement dosing, therefore, Ca(OH)_2_ was the product involved in the primary hydration reaction, while hydrated calcium silicate and calcium alumina (AFt) were the products in the alkali excitation reaction.
(4)$${\mathrm{CSH}}_{1/2}+1.5H\to {\mathrm{CSH}}_{2}$$
(5)$$C_{3}S+5.3H\to C_{1.7}{\mathrm{SH}}_{4}+1.3\mathrm{CH}$$
(6)$$C_{2}S+4.3H\to C_{1.7}{\mathrm{SH}}_{4}+0.3\mathrm{CH}$$
(7)$$C_{3}A+3{\mathrm{CSH}}_{2}+26H\to C_{6}{\mathrm{AS}}_{3}H_{32}$$
(8)$$C_{6}{\mathrm{AS}}_{3}H_{32}+2C_{3}A+4H\to 3C_{4}{\mathrm{ASH}}_{12}$$
(9)$$C_{3}A+6H\to C_{3}{\mathrm{AH}}_{6}$$
(10)$$C_{4}\mathrm{AF}+3{\mathrm{CSH}}_{2}+30H\to C_{6}{\mathrm{AS}}_{3}H_{32}+\mathrm{CH}+{\mathrm{FH}}_{3}$$
(11)$$1.1\mathrm{CH}+S+2.8H\to C_{1.1}{\mathrm{SH}}_{3.9}$$
(12)$$4\mathrm{CH}+A+9H\to C_{4}{\mathrm{AH}}_{13}$$
where CSH is the hydration calcium silicate gel, CH is Ca ${\left(\mathrm{OH}\right)}_{2}$, S is ${\mathrm{SiO}}_{2}$, A is ${\mathrm{Al}}_{2}O_{3},{\;C}_{1.1}{\mathrm{SH}}_{3.9}$ and $C_{4}{\mathrm{AH}}_{13}$ is hydration product.

### 3.3. IR Analyses

The XRD technology can only determine the structure of crystal minerals, and there is no crystal mineral in the product generated by the hydration of CSS at room temperature [23], therefore, information concerning the complete product structure cannot be obtained by XRD technology. As the main structural unit of silicate minerals is [SiO_4_] tetrahedra, the current structure is often characterized by studying the polymerization state of [SiO_4_] tetrahedra in non-crystalline products such as C-S-H gels.

It can be seen from Figure 5 that when the curing age was 7 d, with the gradual increase in the content of CSS, the number of vibration peaks of the corresponding groups at the wavelength of 400–550 cm^−1^ and 800–1000 cm^−1^ were different. When the crystal structure or polymerization degree of the silica tetrahedron changes, the corresponding absorption band generally changes from a low wave number to a high wave number [24,25]. The wavelength of the bending vibration peak of silica-oxygen tetrahedra (400~550 cm^−1^) gradually increases with the increase in silica-calcium slag doping, while the wave number in (983 cm^−1^ and 873 cm^−1^) migrates in the direction of the small wave number. This is because the incorporation of CSS increases the cement of β-C_2_S, while β-C_2_S and its hydration products C-S-H are in a chain-like structure whereby the hydration rate is slow, and at 28 d the unhydrated β-C_2_S still exists. 

The absorption of CSS hydration products in the mid-infrared region is mainly caused by the lattice vibration of the anion, and usually, when the atomic coefficient of the cation increases, the absorption position of the anion group will make a small displacement in the direction of the lower wave number. It has been shown that the symmetric stretching vibration peak of the Si-O bond in the [SiO_4_] tetrahedra of the hydration product C-S-H gel has strong infrared activity at its peak. However, in the C-S-H and C(N)-A-S-H coexistence systems, the characteristic peaks characterizing the Si-O-Si bond stretching vibrations in [SiO_4_] tetrahedra are usually located between wave number 950 and 1000 cm^−1^. Based on the analysis of the above results, partial calcite IR spectra were found in the IR spectra of cement stabilized aggregates at 7 d and 28 d, which was consistent with the results of the XRD analysis.

Figure 5b shows that the [SiO_4_] tetrahedra in C-S-H gels producing hydration products have bending vibration peaks near the wave number 659 cm^−1^, but their intensity is weak. However, with the addition of Al^3+^, C-A-S-H formed in the polymerization stage of C-S-H, which increased the degree of polymerization of [SiO_4_] in the gel and led to the offset of the bending vibration. The area of the bending vibration peaks generated at 1431 cm^−1^ and 874 cm^−1^ increased significantly compared with the curing 7 d which indicated a gradual increase in the polymerization of its Ca(OH)_2_, C-S-H and C-A-S-H, leading to C-O and Si-O bonds, a gradual increase in strength properties on a macroscopic scale.

### 3.4. SEM Analysis

In order to further visualize the hydration products and micromorphology of the cementitious material, SEM analysis was performed on the samples of CFC-1, CFC-2, and CFC-7 after curing for 7 days and 28 days.

The micrographs are shown in Figure 6. The overall compactness was high, and the crystal development was significantly improved. It can be observed in Figure 6A that the hydration products of Ca(OH)_2_ crystals in flakes and of different sizes, fine particles and less fibrous condensed particles were found in the hydration products at the age of 7 d when the CSS was mixed with 13%, and rod-shaped crystals were observed at point g. It can be inferred that the hydration products were mainly C-S-H colloids [26]. At 28 d of hydration age (Figure 6B), the fibrous crystal shape was observed at point h where no pores existed, the Ca(OH)_2_ crystals were basically stacked together according to the plate-like structure, the overall denseness was higher, and the degree of crystal development was significantly enhanced. The hydration products were C-S-H colloids formed by irregular and equally large particles distributed inside the crystal structure and significantly increased in size; in terms of macroscopic properties, their compressive strength was significantly increased. As the age of hydration increased, the content of hydration products in the material increased significantly.

With respect to Figure 6C, it is understood that when the CSS amount was 11% hydration age 7 d, the hydrated product was a large particulate body formed by the C-S-H colloid and the adjoined fibrous gel observed at point i The particles illustrate that the elongated strip substances had grown from cement particles early, in water. It may be observed in Figure 6A, that the structure has a large gap and a gel substance around the C point, and a small amount of fibrous gel-type particles are observed, indicating that a C-A-S-H is consistent with the results of XRD graph analysis. Compared with 7 d age, there are still a large number of large particles in the 28 d population, but the size of these products is smaller, more intimate, and the sample overall is obviously more tight. This indicates that the potential active components, such as β-C_2_S in CSS, gradually participate in chemical excitation reaction with the extension of age [23]. By observation of Figure 6D, the size of fibrous gel particles in j region is obviously larger, and the diffraction peak intensity of AFt in XRD diagram is obviously enhanced, indicating that the polymerization degree of C-A-S-H gel has gradually increased and the overall structure is more dense, which is the direct reason for the increase in the strength of the sample with the extension of the age.

In Figure 6A it can be seen that when the CSS dosage was 0% at curing 7 d, the generated CSH colloid was significantly reduced, compared with the Ca(OH)_2_ gel. The area of was significantly reduced, and structurally, Ca(OH)_2_ and C-S-H colloidal distributions were not close, indicating that the incorporation of powder coal ash was reduced, the hydration of the ceiling filling was reduced and the activity of SiO_2_ and Al_2_O_3_ could not be completed. Henceforth, the internal tightness of the sample was reduced, the internal pore in the overall sample was increased, and the dense structure could not be reached, resulting in the decrease in the early compressive strength of the gelation material as the amount of CSS dumping was reduced. As the raging age increased, the secondary hydration reaction of SiO_2_ and Al_2_O_3_ in the inorganic binding material in the inorganic binding material increased, and the number of Swed-shaped C-S-H was significantly increased in the secondary hydration reaction of SiO_2_ and Al_2_O_3_ in the inorganic binding material. The wrapped-in-the-surface structure of the FA became tight, preventing a large number of OH, the active ingredients in the FA, from participating again in the hydration reaction, thereby reducing its hydration reaction. At point k, the material formed into a rod-like AFt, indicating that the Ca/Si content in the material was low, and the Ca^2+^ content was low [23], and the residual [SiO_4_] could not continue the polymerization reaction to form a net-like compact structure.

### 3.5. Freeze Analysis

The wide application of road substrates requires consideration not only of their mechanical properties but also of their durability, which must therefore be analyzed. A focus on the analysis of frost resistance and water stability of road base materials follows.

#### 3.5.1. Freeze–Thaw Testing

As the UCS of CFC-2, CFC-3, CFC-4, and CFC-7 were in accordance with the technical standards of road subgrade, the focus is on freeze–thaw and durability tests for these four groups. According to Chinese standard JTG E51-2009, the UCS before and after 5 and 10 freeze–thaws were used to analyze the freeze stability of the materials. The freeze–thaw test was carried out at 28 d of age and the UCS after 5 freeze–thaws was compared with that before freeze–thaw. The freeze–thaw test was also carried out at 180 d of age and the UCS after 10 freeze–thaws was compared with that before freeze–thaw to analyze the freeze–stability of the material. The specimens were perked under standard permaculture conditions, and on the last day of the oxygenation period the specimens were taken out of the maintenance room, and were immersed in a water bath with the water surface higher than 2.5 cm above the specimens. If the mass loss rate exceeded 5% during the experiment, the freeze–thaw cycle could be stopped, and the freeze resistance index was calculated according to Equations (13) and (14).
(13)$$\mathrm{BDR}=\frac{\mathrm{RDC}}{\mathrm{RC}}\times 100$$
(14)$$W_{n}=\frac{m_{0}-m_{n}}{m_{0}}\times 100$$
where BDR is the intensity loss rate of n times $\;m_{0}$, RDC is the strength after n times of freezing and thawing, RC is the compressive strength of the comparison specimen, W_n_ is the rate of mass change after n freeze–thaw cycles, $m_{0}$ is the mass before the freeze–thaw cycle, and $m_{n}$ is the mass after n freeze–thaw cycles.

According to the calculation results of Equations (13) and (14), as shown in Figure 7, the loss rate of UCS of CFC-2, CFC-3, CFC-4 and CFC-7 after 5 freeze–thaw cycles were 5.32%, 5.31%, 4.52%, and 4.36%, respectively, and the unconfined compressive strength after 10 freeze–thaw cycles were 7.62%, 7.98%, 6.5%, and 6.23%, respectively, with residual strengths greater than 90%. The UCS after 5 freeze–thaw cycles was significantly smaller than the unconfined compressive strength after 10 freeze–thaw cycles, which is in accordance with the law of pavement freeze–thaw performance development. The CFC-2, CFC-3, CFC-4, and CFC-7 still had high compressive strength after 5 and 10 freeze–thaws. Figure 7b shows that the mass loss rate of the specimens becomes smaller as the amount of CSS is increased, indicating that the incorporation of calcium silica slag was beneficial to the freeze–thaw resistance of the mix. This also indicated the feasibility of CSS preparation for road base materials.

#### 3.5.2. Water Stability

The UCS test is divided into two cases of water immersion and non-immersion. In order to investigate the water stability of CSS stabilized macadam material, CSS was carried out according to two ages and two cases of immersion and non-immersion. The samples were immersed in water one day before the experiment, and then the experiment was carried out. The water stability factor is calculated as follows (15):(15)$$\mathrm{WSC}=\frac{M_{1}}{M_{2}}\times 100$$
where WSC is the water stability coefficient (%), M_1_ is UCS after n-day immersion, and M_2_ is the UCS of n-days without immersion.

According to the calculation results of Equation (15), as shown in Figure 8, the water stability coefficients of samples CFC-2, CFC-3, CFC-4 and CFC7 were 67.2%, 68.9%, 72.4% and 73.1% at 7 d, and 71.5%, 72.4%, 74% and 75.5% at 28 d, respectively. The water stability of CSS aggregates met the technical requirements of Chinese subgrade. As can be seen from Figure 9, the water stability coefficient increased relatively with the growth of age and the increase in the mass proportion of CSS, which also indicated that CSS helped to enhance the water stability of the base material.

### 3.6. Field Application

A pilot trial of road base material prepared from CSS has been completed in Hohhot, Inner Mongolia, China. The pictures of the pilot site are shown in Figure 9. The UCS of CFC-2 prepared from CSS were tested at 7 d, 14 d, 28 d, 90 d and 180 d, as 5.5 MPa, 7.3 MPa, 8.7 MPa, 12.8 MPa and 13.9 MPa, respectively. These results are in accordance with the Chinese standard specifications, and the frost resistance and water stability tests are in accordance with the technical standards of the subgrade. Figure 9 shows the CSS pavement subgrade drill core sampling. The regular core sample shape is flat and smooth, indicating good integrity, and that the CSS, fly ash and coarse aggregate cementation is good. This shows that CSS is feasible in the application of the preparation of road base materials, and can replace traditional cement lime road base materials.

## 4. Conclusions

In this paper, calcium silicate slag is used as the main material for the preparation of inorganic binding materials for pavement subgrade, and mechanical properties, durability, mineralogical phase, and microscopic morphology were studied for characterization. The following conclusions can be outlined as follows:
(1)The peak UCS of pavement base material (CFC-2) prepared by CSS:FA:cement of 11:2:2 at 7 d can reach 5.5 MPa, which is in accordance with the Chinese road standard (3–5 MPa). The unconfined compressive strength at 90 d and 180 d can reach 8.7 MPa and 13.9 MPa, and the durability of the road base material also meets the standard requirements;(2)The appropriate amount of CSS incorporation improved the UCS and water stability coefficient of cement stabilized materials, and reduced the mass loss rate of freeze–thaw cycles, indicating that the appropriate amount of CSS incorporation can improve the mechanical properties and durability of substrate materials;(3)As shown by XRD and FTIR results, with the increase in CSS doping in inorganic bonding materials, the generated hydration products C-S-H gel and C(N)-A-S-H gel gradually increased, and the degree of [SiO_4_] polymerization in the hydration products increased, which promoted the degree of hydration of inorganic bonding materials;(4)SEM result analysis showed that due to the increase in CSS content, the needle type structure of hydrated calcium silicate aluminate and gel hydrated calcium silicate transformed into columnar structure and the subsequent crystal structure is more set dense;(5)The pilot test confirmed the feasibility of CSS as a road subgrade material. Compared with cement-fly ash road base material, the incorporation of CSS significantly improved the mechanical properties and durability of road base material. CSS road base material not only replaces expensive traditional materials with secondary solid waste generated by fly ash extraction technology, which reduces the engineering cost of road base material, but also consumes a large amount of industrial solid waste, creating a green cycle of resources and protecting the ecological environment.


## Figures and Tables

**Figure 1 materials-15-02304-f001:**
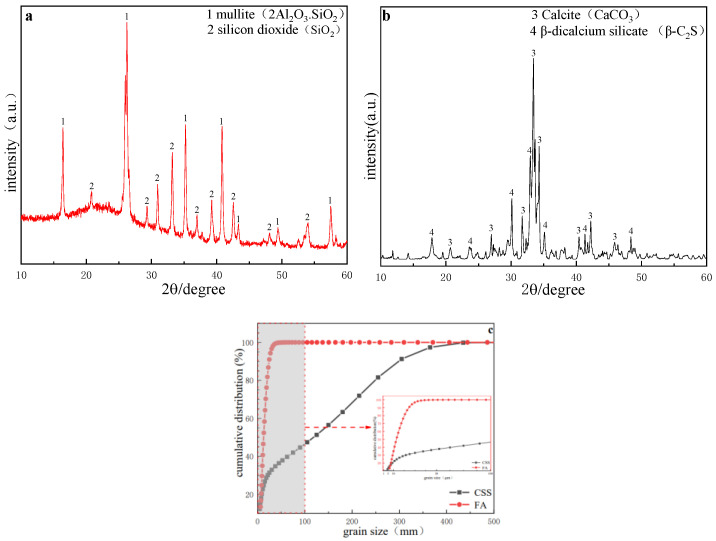
Characteristics of raw materials: (**a**) XRD pattern of CSS, (**b**) XRD pattern of FA, (**c**) size distribution of CSS and FA.

**Figure 2 materials-15-02304-f002:**
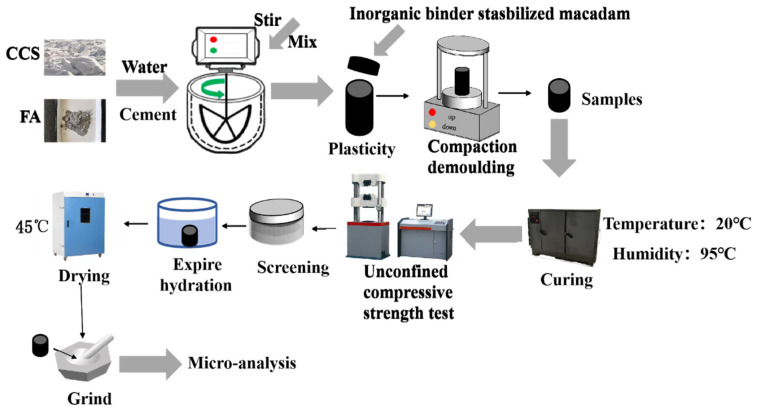
Flow chart for the testing of CSS pavement base materials.

**Figure 3 materials-15-02304-f003:**
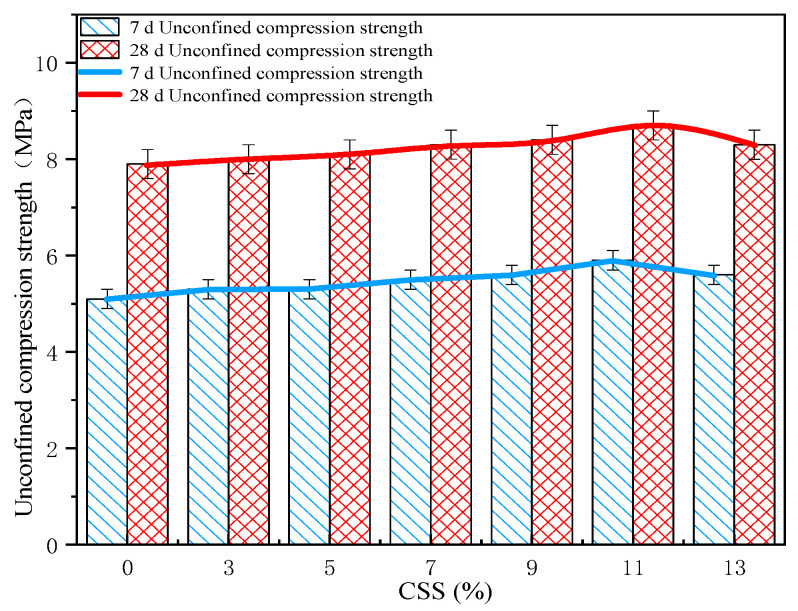
Different proportions of UCS at 7 d and 28 d.

**Figure 4 materials-15-02304-f004:**
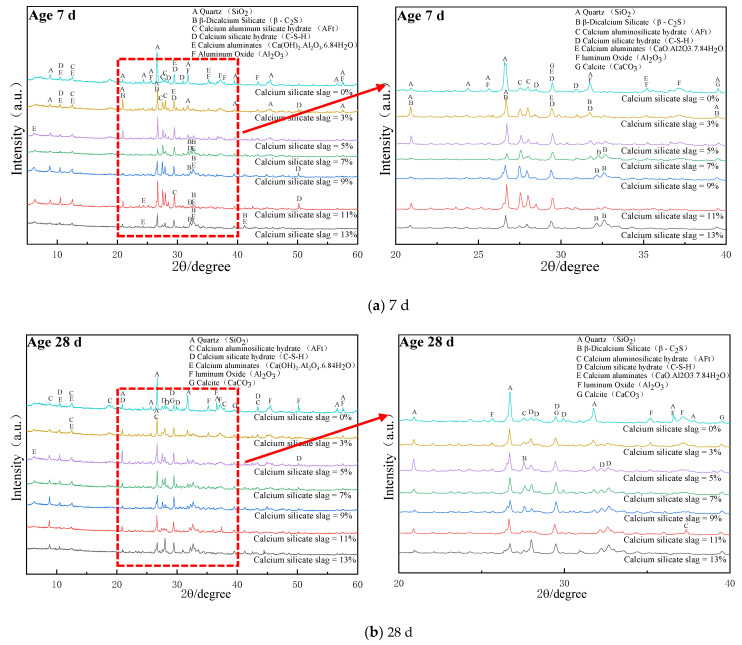
XRD patterns under different dosages of calcium silicate slag: (**a**) XRD analysis of curing for 7 d, (**b**) XRD analysis of curing for 28 d.

**Figure 5 materials-15-02304-f005:**
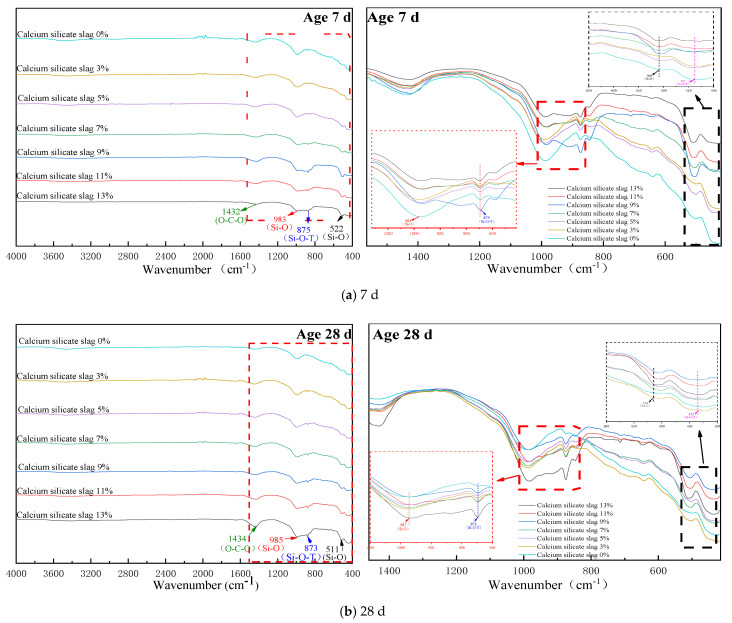
CSS base layer material FTIR diagram: (**a**) FTIR analysis of curing for 7 d, (**b**) FTIR analysis of curing for 28 d.

**Figure 6 materials-15-02304-f006:**
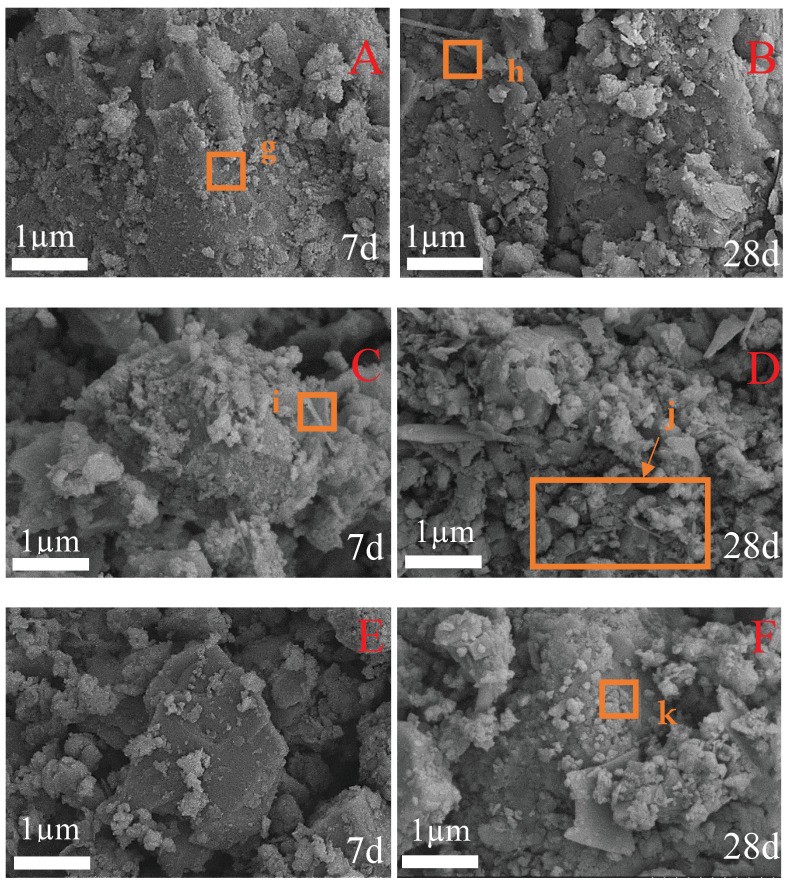
SEM chart of subgrade materials for CSS base pavement: (**A**) CFC-1 cured 7 d, (**B**) CFC-2 cured 28 d, (**C**) CFC-2 cured 7 d, (**D**) CFC-2 cured 28 d, (**E**) CFC-7 cured 7 d, (**F**) CFC-7 cured 28 d.

**Figure 7 materials-15-02304-f007:**
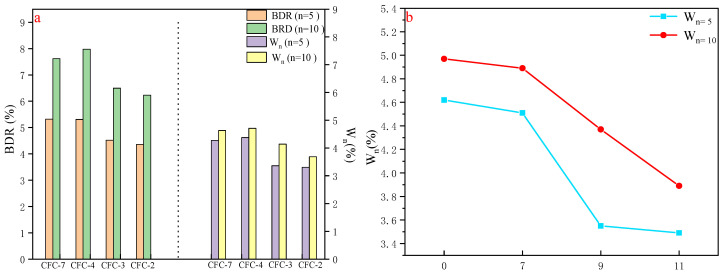
Freeze–thaw cycle test results (**a**) comparison of freeze–thaw loss rates, (**b**) mass loss rates after n freeze–thaws for different CSS.

**Figure 8 materials-15-02304-f008:**
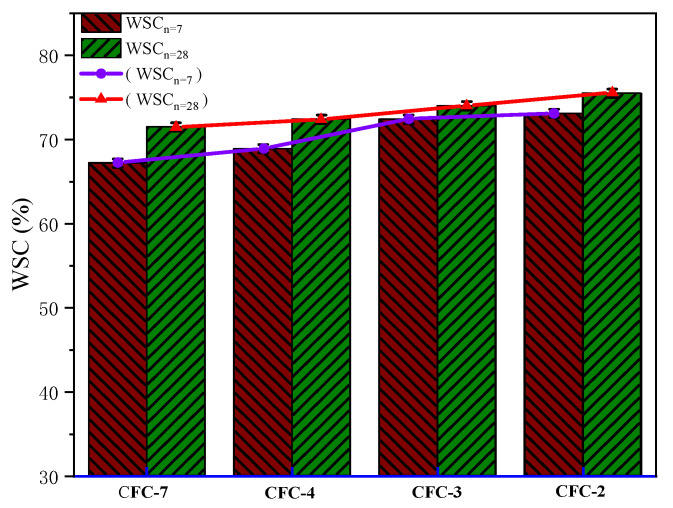
Water stability coefficient of calcium silicate slag stabilized base material.

**Figure 9 materials-15-02304-f009:**
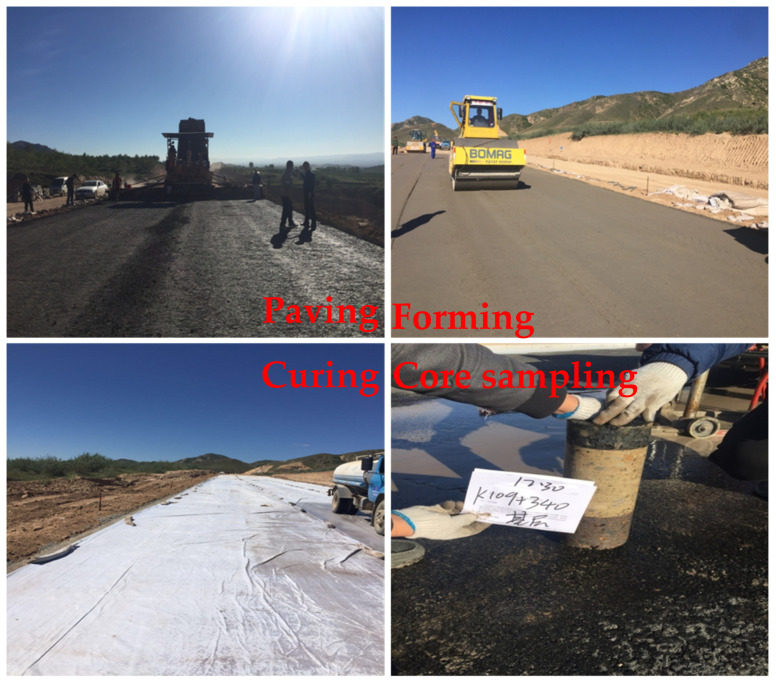
Field picture of the pilot test.

**Table 1 materials-15-02304-t001:** Chemical composition of CSS, FA (wt.%).

Chemical Composition	SiO_2_	Fe_2_O_3_	Al_2_O_3_	CaO	MgO	Na_2_O	K_2_O	LOI
CSS	31.08	2.25	5.97	50.35	3.61	2.31	0.36	4.07
FA	42.67	2.57	42.36	4.30	3.20	0.58	0.39	3.93

**Table 2 materials-15-02304-t002:** Test scheme of calcium silicate slag pavement base material (wt.%).

Number	CSS	FA	Cement
CFC-1	13	0	2
CFC-2	11	2	2
CFC-3	9	4	2
CFC-4	7	6	2
CFC-5	5	6	2
CFC-6	3	10	2
CFC-7	0	13	2

**Table 3 materials-15-02304-t003:** Information on mineral phases of hydration synthesis of different CSS pavement base materials.

No.	Phase	Chemical Formula	PDF Card No.	Main 2θ (degree)
A	Quartz	SiO_2_	01-085-0930	26.657, 20.876, 50.143
B	β-dicalcium silicate	β-2CaO·SiO_2_	01-083-0465	32.168, 32.599, 41.275
C	AFt	C(N)-A-S-H	00-039-0217	12.439, 28.037, 29.524
D	C-S-H	C-S-H	00-042-0538	29.454, 30.452, 50.058
E	Calcium aluminum hydroxide hydrate	Ca(OH)_2_·Al_2_O_3_·6.84H_2_O	01-088-1410	6.223, 12.465, 24.218
F	Aluminum oxide	Al_2_O_3_	01-078-2427	25.585, 35.134, 43.359
G	Calcite	CaCO_3_	010-083-1762	29.410, 36.045, 39.401

## Data Availability

Not applicable.

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
