# Peer review of "Experimental Study on Subgrade Material of Calcium Silicate Slag"

_materials, 2022, doi:10.3390/ma15062304_

Round 1
Reviewer 1 Report
The paper is titled “Experimental study on subgrade material of calcium silicon slag base pavement".
I recommend Major revision of this paper prior to be publish in Materials journal.
Some comments are provided below that could be helpful for the authors to improve their manuscript.
- The work evaluated requires a thorough review by the authors.
- The introduction does not show that the proposed objectives are new or contribute new knowledge to the topic under study. In addition, it should be increased for 2 pages max.
- Correct/ calcium silicate slag (CCS) to be CSS in all manuscript.
- In Table 2 Test scheme of silicon calcium slag pavement base material, indicate your mix composition done in weight or percent.
- You should mention the surface area for both CSS and FA, and they are compatible with cement surface area.
- No need for the experiment figure in the manuscript as there are no physical properties for asphalt are performs, such as temperature sweep and penetration (100gm.5sec.25℃). In addition, you should compare your results with reference pavement sample.
- Figures and tables have to be revised and re-arranged
Author Response
Thank you very much for your comments on our paper in your busy schedule,We have revised our paper according to your comments,please see the attachment,

Reviewer 2 Report
General impression#
The valorization of coal-combustion fly ash, with addition of inorganic binder, is widely applicable in pavement construction. The topic of presented research is multi-applicable in various fields of engineering: chemical, civil and environmental, including material science, as well. That is the main benefit of presented research. Otherwise, the manuscript possesses many fundamental and technical obstacles and omissions. The structure is not well organized and text flow is not easy to follow (expect the part related to structural analyses – XRD, IR and FIR; those are good textual parts). Before the consideration to be published, the manuscript needs profound and detailed improvement:
Comment#1. The whole manuscript should be peer-reviewed by native English person.
Comment#2. The whole manuscript should be technically improved:
- Spaces (almost everywhere in the text; especially where the units are expressed and/or the brackets of references).
- Units (units should be written separately from the numerical value; except in the case of ’%’)
- Fonts (there are several different fonts; for exp. Page 3, Line 92. Experimental method)
- References used in the text are in brackets. Please use the ’.’ after ’[]’. There are several mistakes; for exp. Page 2, line 67.
- The captions of Figure 2, Figure 4 and Figure 5 are missing.
- The caption of Figure 3 has to be corrected: Unconfined compressive strength of silicon calcium slag pavement base material.
- All equations should be typed in Equation Editor (page 5 and page 6).
- Key words should be capitalized: Calcium silicate slag; Inorganic binding materials, Construction material, Pavement base, Micromorphology.
The most probably, the list of technical corrections is not final, but for this stage of manuscript evaluation is mandatory.
Comment#3. The title has to be rephrased: Mechanical properties of subgrade material of calcium-silicate slag base pavement, in order to express the real findings of research.
Comment#4. The Abstract should be rewritten – its hard to follow the text flow. Furthermore, the first paragraph in Abstract has to be removed (line 27-35).
Comment#5. The XRD analysis was performed for powder samples? Please provide the cards/patterns of the main detected constituents of the sample.
Comment#6. Conclusion has to be written in a clear and more understandable way. For instance, the sentence: ’The incorporation of silicon calcid is increased the limited comprehensive strength of cement stabilizing material....', is not clear, or even finished. Please rephrase.
Further, the third point (3) in Conclusion is also hard to understand (in the same sentence there are several comparisons using the same context – something is increasing, something is decreasing..). Please rephrase or split the sentence into two parts. The final messages (outcomes) have to be punctual, clear and short.
Comment#7. Please explain the influence of hydratation factor and highlight its importance on mechanical properties of tested material, there are opposite conclusions related to this factor in the manuscript.
Author Response
Thank you very much for your comments on our paper in your busy schedule, We have revised our paper according to your comments.please see the attachment.

Reviewer 3 Report
The overall manuscript should be revised. There is no proper connection and difficult to understand.
Hence, it is recommended the author's to develop the manuscript proper understanding to readable.
- The overall manuscript looks unfair. The authors submitted the manuscript without any concern / care and without proper framing.
- In Introduction part "The introduction should briefly..................details on references". This paragraph does not make any sense. It looks like from any previous journal suggestion / comment from reviewer's (or) work supervisor suggestions. The authors pasted as it is like reply to comments.
- The font size and font type should be consider.
- The Introduction, novelty, motivation and methodology of the work is not clear. Authors should focus on them.
- The Author's should maintain proper connections in the sentences and suggested to frame sentences with any English professional /academic professional.
- There is no Figure numbers and titles for the figures 2, 4(XRD), 5 (FTIR). Then how authors have cited them.
- There should be proper connection of explanation / discussions for those figures.
- The authors have to reconsider the conclusion part. Frame the Abstract by consideration of whole work.
Author Response
Thank you very much for your comments on our paper in your busy schedule, We have revised our paper according to your comments, please see the attachment.

Reviewer 4 Report
The manuscript presented a well-organized research paper. However, following comments are advised to be considered before acceptance:
- Abstract
Abstract is well written and ok.
- Introduction
Technical issues in introduction parts were observed. For example, line 36 as we all know. Line 43 what is FA In last paragraph of introduction, the author mention there was few studies, where is few studies. Please little explain these studies and what is difference in theses past studies and these studies. Furthermore, the novelty of this study is not ok
- Experimental
Why CaO is too high in CSS. Please provide any reference which support this.
Why compressive strength is decreased at 13% of CSS. The authors did not mention the reason for increased and decreased strength. Furthermore, also compare these results with past study. Also mention the reference for maximum strength at 11% of CSS.
3.4 Spelling of analysis
4.Conclusion
Analysis of SEM results, with the increase of silicon-plavous dumping, the hydrated 352 silicate content of the needle type is gradually increased, and the structure is more preferable, 353 which helps improve Mechanical properties and durability of pavement base materials.
Where is durability. It suggested to do some durability tests. Also mention the drawback of this study.
Overall, I do not recommend publication of this article in current form. Please carefully look the spelling. I observed in many places. Some of them I mention. Furthermore, some formatting issues like table 2 etc.
Author Response
Thank you very much for your comments on our paper in your busy schedule, We have revised our paper according to your comments.
Regarding the response to your comment, please check the attachment.

Reviewer 5 Report
Dear Author,
Comments are Appended

Author Response
Thank you Very much for your comments on our paper in your busy schedule, we have revised our paper according to your comments.
Please check the attachment.

Round 2
Reviewer 1 Report
Dear authors
Good job, you did and I wish you all the best.
Best Rgards
Reviewer 2 Report
Manuscript#materials-1625722
General impression: The authors have accepted reviewers' suggestions (including additional amendments) and the overall quality of the manuscript is significantly improved. The real-scale application of presented research has the most significant outcome for the readers, and further on, wider scientific auditorium.
Reviewer 3 Report
It would be easier to review and understand, if the authors could have outlined / highlighted only the revised parts of the manuscript instead of strike out the sentences and revised them in same red colour.
However, the manuscript can be accepted by removing the strike out parts and submit revised manuscript by outlining / highlighting the only revised parts of the manuscript.
Reviewer 4 Report
The authors addressed my comments.